# Ego-Vehicle Speed Correction for Automotive Radar Systems Using Convolutional Neural Networks

**DOI:** 10.3390/s24196409

**Published:** 2024-10-03

**Authors:** Sunghoon Moon, Daehyun Kim, Younglok Kim

**Affiliations:** Department of Electronic Engineering, Sogang University, Seoul 04107, Republic of Korea; shmoon@sogang.ac.kr (S.M.); sinzuggo@sogang.ac.kr (D.K.)

**Keywords:** automotive radar system, ego-vehicle speed, convolutional neural network, vehicle speed correction, speed ratio, deep learning

## Abstract

The development of autonomous driving vehicles has increased the global demand for robust and efficient automotive radar systems. This study proposes an automotive radar-based ego-vehicle speed detection network (AVSD Net) model using convolutional neural networks for estimating the speed of the ego vehicle. The preprocessing and postprocessing methods used for vehicle speed correction are presented in detail. The AVSD Net model exhibits characteristics that are independent of the angular performance of the radar system and its mounting angle on the vehicle, thereby reducing the loss of the maximum detection range without requiring a downward or wide beam for the elevation angle. The ego-vehicle speed is effectively estimated when the range–velocity spectrum data are input into the model. Moreover, preprocessing and postprocessing facilitate an accurate correction of the ego-vehicle speed while reducing the complexity of the model, enabling its application to embedded systems. The proposed ego-vehicle speed correction method can improve safety in various applications, such as autonomous emergency braking systems, forward collision avoidance assist, adaptive cruise control, rear cross-traffic alert, and blind spot detection systems.

## 1. Introduction

The advancement of autonomous driving vehicles and the mandatory implementation of autonomous emergency braking systems (AEBSs) in various countries have increased the demand for automotive radar systems that offer increased robustness to adverse weather and variable lighting conditions while providing long-range and accurate measurements [1,2,3,4]. In addition to their application to AEBSs, automotive radar systems can be used in various safety applications, such as forward collision-avoidance assist (FCA), adaptive cruise control (ACC), rear cross-traffic alert, and blind spot detection systems [5,6]. Unlike other sensors—such as cameras and lidar—used in autonomous driving systems, reliable radar systems that can better withstand weather conditions such as sunlight and fog and provide accurate measurements underlie the continued advancement of autonomous driving technology, thereby expanding the scope of applications of radar technology in autonomous vehicles.

Determining the absolute speed of an object is crucial in automotive radar systems. These systems differentiate between the moving and stationary states of an object at absolute object speeds [7]. This distinction is crucial for decision-making during ACC when onward vehicles are being tracked and for AEBS and FCA systems for predicting the time to collision and providing information for emergency braking and collision-warning alarms [8,9]. Additionally, the distinction between moving and stationary states facilitates the classification of objects such as vehicles, bicycles, and pedestrians [10] and aids in determining and removing clutter, which reduces radar detection performance [11,12].

In radar systems that can only measure the relative speed of objects, the absolute speed of an object is determined by combining the ego-vehicle speed and the relative speed of the object detected by the radar. Therefore, accurate information on the speed of the ego-vehicle must be obtained. Generally, automotive radar systems receive minimal vehicle data, such as the wheel speed and yaw rate, from the Controller Area Network (CAN). The ego-vehicle speed is indirectly measured via the wheel-rotation speed monitored using a wheel-speed sensor [13,14,15]. However, if the contact area between the tire and road increases because of the air pressure or wear of the tire, the radius of the tire may decrease, resulting in a deviation in the ego-vehicle speed estimation [16,17,18]. Therefore, the ego-vehicle speed information in automotive radar systems must be corrected to minimize the object speed distortion caused by this deviation.

To this end, we propose an automotive radar-based ego-vehicle speed detection network (AVSD Net) model that uses convolutional neural networks (CNNs) and outputs the encoding value of the ego-vehicle speed feature from the range-velocity (RV) spectrum. Furthermore, we introduce postprocessing to increase the accuracy of the ego-vehicle speed extracted using the AVSD Net model. The primary contributions of the study can be summarized as follows:The proposed model is independent of the radar mounting angle on the vehicle.The model is independent of the angular performance of the automotive radar system.The loss in the maximum detection distance is reduced because the model is not operated with a downward or wide beam for the elevation angle.The complexity is reduced by simplifying the model and reducing the parameters, enabling the application of the model in embedded systems.

The remainder of this paper is organized as follows: Section 2 reviews related research on ego-vehicle speed correction. Section 3 introduces the developed method and presents the results of the measurement and simulation data. The proposed CNN-based model used to estimate the ego-vehicle speed and the preprocessing and postprocessing steps are also explained in this section. Section 4 discusses the results of the AVSD Net model and analyzes the postprocessing results described in Section 3. Finally, Section 5 summarizes the conclusions of the study.

## 2. Related Work

Several methods have been proposed for correcting the ego-vehicle speed.

Orthogonal distance regression (ODR): In a previous study [18], ODR algorithms were used to analyze the correlation between the speed of an ego vehicle and the relative speed/angle of stationary objects to predict the ego-vehicle speed. When there was significant noise in the measured values or when stationary objects were not detected, the ego-vehicle speed was predicted using the Kalman filter.Hough transform: Lim and Lee [19] aimed to accurately estimate the speed of the ego vehicle by examining the distribution characteristics of stationary objects within an angle-velocity domain and by identifying the intersection point of the Hough transform.Elevated objects: Kingery and Song [20] proposed a method to compensate for the elevation angle and estimate the ego-vehicle speed by analyzing the radial velocity difference in elevated objects, such as buildings and tall trees.Ground backscattering: A unique technique has been proposed [21] for estimating the ego-vehicle speed by analyzing backscattering signals reflected from the ground.Random sample consensus (RANSAC): The speed of the ego-vehicle was matched by distinguishing stationary objects using the RANSAC algorithm and interpreting the radial velocity errors associated with these stationary objects [22].Complex value neural network (CVNN): An innovative approach presented in [23] and [24] provides a method for predicting the ego-vehicle speed using a sophisticated deep network model, referred to as CVNN, wherein the multi-channel RV spectrum serves as input data for the model.

In previous studies [18,19,20,21], the angular performance of stationary objects was an important parameter that relied significantly on the angular accuracy of the radar system and its mounting angle on the vehicle. In [22], the yaw rate was separately estimated to keep it independent of the radar mounting angle; however, the estimation of ego-vehicle speed was dependent on the radar angle performance. Additionally, certain studies required intentional beam steering or a wide beam at an elevated angle [20,21], which reduced the maximum detection range. Although specifics regarding the number of layers and parameters in the deep network were omitted in [23,24], the network architecture comprised a series of deep feature extractions composed of multiple residual groups, which in turn included multiple residual blocks. Moreover, the utilization of complex numbers increased the complexity of these methods.

Table 1 summarizes the constraints associated with each existing method used for estimating the ego-vehicle speed.

This study presents a novel model for predicting and correcting the ego-vehicle speed for the automotive radar system using a CNN to overcome the constraints reported in previous studies. The developed method is independent of the angular performance of the radar and its mounting angle on the vehicle. This reduces the loss of the maximum detection range because a downward or wide beam is not required for the elevation angle. Furthermore, the proposed model can be applied to embedded systems owing to its simplicity and reduced parameter size.

## 3. Materials and Methods

### 3.1. Measurement Dataset

#### 3.1.1. Configuration of the Vehicle and Radar System

We used the automotive radar system built by Texas Instruments (TI) with an AWR2944 board, which is a millimeter wave (mmWave) sensor developed for automotive radar system applications that operate within the frequency range of 76–81 GHz [25]. Subsequently, a sample from an analog-to-digital conversion (ADC) was obtained by interfacing it with a TI DCA1000 [26]. A global positioning system (GPS) module was used to obtain reference information for the ego-vehicle speed and heading angle. The ego-vehicle speed was extracted from the CAN bus based on the wheel speed. Driving videos were recorded using a webcam to monitor the driving conditions. Figure 1 illustrates the configurations of the vehicle and radar system.

#### 3.1.2. Waveform Specifications of the Automotive Radar System

Figure 2 depicts the waveform of the automotive radar system used in this study. Table 2 details the waveform specifications.

The waveform set exhibited a frequency of 77.1–77.31 GHz with a bandwidth (BW) of 211.35 MHz, sample rate (SR) of 20 MHz, and an ADC sampling time ($T_{ADC}$) of 12.8 µs. The range resolution ($R_{res}$) was determined to be 0.71 m using Equation (1) [27,28].
(1)$$R_{res}=\frac{c}{2\;BW},$$
where *c* denotes the speed of light. The maximum detection range ($R_{max}$) was 90.78 m and was determined using Equation (2) [27,28] as follows:(2)$$R_{max}=\frac{c\cdot T_{ADC}\cdot SR}{4\;BW}.$$

The wavelength (λ) for the center frequency ($f_{c}$) of 77.2 GHz in the frequency range of 77.1–77.31 GHz is 3.883 mm, and the velocity resolution ($V_{res}$) 0.38 m/s is defined using Equation (3) [27,28].
(3)$$V_{res}=\frac{c}{2\;f_{c}\cdot PRI\cdot N_{c}\;}=\frac{\lambda }{2\cdot PRI\cdot N_{c}}.$$
The maximum velocity ($V_{max}$) is 24.27 m/s, which can be expressed as follows [27,28]:(4)$$V_{max}=\frac{c}{4\;f_{c}\cdot PRI}=\frac{\lambda }{4\cdot PRI}.$$

#### 3.1.3. Measurement Data Results

We measured the data twice to obtain datasets of 28 min 46 s and 32 min 26 s. Figure 3 presents the measurement results at an ego-vehicle speed of 17.5 m/s. The dashed and solid red lines in each figure represent the stationary objects on the left- and right-hand sides, which are the left and right guardrails, respectively.

This result indicates that the radial velocity is affected by the angle of the object when stationary objects are close to the automotive radar. Conversely, when stationary objects are far from the automotive radar, the radial velocity is less affected by the angle of the objects. For instance, the angle of the stationary object at 90 m tends to approach 0°, indicating that the radial velocity of the stationary object approaches −17.5 m/s, which is the opposite of the ego-vehicle speed. As illustrated in Figure 4, when a vehicle equipped with an automotive radar moves in the y direction, the angle of the green stationary object is larger than that of the blue one; therefore, the radial velocity relatively changes more. The red stationary object placed at an extremely long distance has a considerably small angle; therefore, the vehicle receives a speed that is close to the opposite of the ego-vehicle speed. The relationship between the stationary target angle (θ), radial velocity ($V_{r}$), and ego-vehicle speed ($V_{ego}$) can be presented as follows [18,20]:(5)$$V_{r}=-V_{ego}\cdot cos\left(\theta \right).$$

Figure 4b depicts the radial velocity trend with respect to position x. We observed that the velocity is close to the opposite speed of the ego vehicle at approximately 60 m or higher. We intend to estimate the ego-vehicle speed by training features where distant stationary objects converge to a velocity opposite to that of the ego vehicle, as depicted in Figure 3b.

As explained in Section 1, the ego-vehicle speed of the CAN signal based on the wheel speed may differ from the actual speed owing to differences in the tire radius caused by the air pressure and wear of the tire. Figure 5 illustrates the ego-vehicle speed measured by the GPS and CAN. Considering the ego-vehicle speed obtained from the GPS signal as reference data, the speed ratio was calculated as an indicator of the difference in the ego-vehicle speed of the wheel-speed-based CAN signal, as follows:(6)$$Speed\;Ratio=median\left(\;\frac{{\left[Egovehicle\;Speed\;from\;GPS\right]}_{i}}{{\left[Egovehicle\;Speed\;from\;CAN\right]}_{i}}\;\right),$$
where *i* represents the sequentially collected data. However, the GPS signal may not be received, and errors may occur because of non-line-of-sight conditions caused by tunnels or high-rise buildings, which may reduce the accuracy of the measured ego-vehicle speed [29,30]. Moreover, in the case of CAN signals based on wheel-speed sensors, ego-vehicle speed errors may occur when the vehicle is cornering [31]. Therefore, the speed ratio was calculated as the median value to minimize errors induced by the outliers in the GPS and CAN data. However, based on the measured results, the calculated speed ratio is 1.0183, indicating a difference of 1.8%. This discrepancy demonstrates the necessity of precisely correcting the ego-vehicle speed in automotive radar systems to collect accurate and reliable information before defining the object properties.

It is crucial to consider several aspects to determine whether only the measured results can be used for training. First, we acquired two datasets of 28 min 46 s and 32 min 26 s using the automotive radar, each comprising 34,538 and 38,909 frames, respectively. A sufficient amount of data appears to be available when the complete dataset is considered. However, if the outlier data generated by GPS failure and cornering are excluded, limited usable data remain. Second, the automotive radar data measured at intervals of 0.5 ms are likely to overfit during training because of the continuous data. For instance, the automotive radar data at a specific time (t) are not significantly different from the data at the previous instant of time (t − 1) and the subsequent instant of time (t + 1). Third, data diversity may not be secured based on the driving habits of the driver. For instance, data obtained from drivers who prefer driving in a specific lane or at high or low speeds are not diverse. Consequently, data diversity must be ensured via simulated data to enhance the training results.

### 3.2. Simulation Dataset

#### 3.2.1. Simulation Methods

The simulation was performed under the same conditions as the automotive radar system used for the measurements. Four channels were used for Tx and Rx, and the antenna channel spacing and pattern were simulated by referring to the antenna pattern image included in the TI AWR2944 user guide [32]. Furthermore, the waveform used in the simulation corresponds to the specifications presented in Table 2.

When the target information is predetermined (Figure 6), the distance between the target and Tx (${\left[R_{tx}\right]}_{n}$) can be calculated based on their corresponding positions, and the distance between the target and Rx (${\left[R_{rx}\right]}_{m}$) can be obtained from the positions of the target and Rx, respectively. Subsequently, the distance between the target and each TRx channel (${\left[R_{trx}\right]}_{n,m}$) can be derived as follows:(7)$${\left[R_{trx}\right]}_{n,m}={\left[R_{tx}\right]}_{n}+{\left[R_{rx}\right]}_{m}.$$

The radial velocity ($V_{r}$) of the target can be defined based on the predefined velocity ($V_{t}$) and angle ($\theta_{t}$) of the target and ego-vehicle speed ($V_{ego}$), as follows:(8)$$V_{r}=\left(V_{t}-V_{ego}\right)\cdot cos\left(\theta_{t}\right).$$
The frequency of the frequency-modulated continuous wave radar can be expressed as
(9)$$f\left(t\right)=f_{0}+∆ft,$$
where $f_{0}$ represents the start frequency and ∆ft denotes the modulated frequency over time. The transmitter signal can be expressed as
(10)$$S_{tx}\left(t\right)=Acos\left[2\pi f\left(t\right)t\right],$$
where *A* denotes the amplitude. Equation (10) can be modified using Equation (9) as follows:(11)$$S_{tx}\left(t\right)=Acos\left[2\pi \left(f_{0}t+∆ft^{2}\right)\right].$$
Considering the distance (R) and radial velocity ($V_{r}$) of the target, the delay time (τ) can be calculated as
(12)$$\tau =\frac{2(R+V_{r}t)}{c}.$$
The received signal can then be expressed as follows:(13)$$S_{rx}\left(t\right)=c\mathrm{os}\left[2\pi \left(f_{0}\left(t-\tau \right)+∆f{\left(t-\tau \right)}^{2}\right)\right].$$
Finally, the ADC signal can be simulated using the mixed signal as
(14)$$S_{mix}\left(t\right)=S_{tx}\left(t\right)\cdot S_{rx}\left(t\right).$$
If the previously described equations are expressed in a more algorithmic form, with subscript $s_{i}$ as the set of the number of ADC samples ($N_{s}$) and subscript $c_{i}$ as the set of the number of chirps ($N_{c}$), we obtain
(15)$$s_{i}=\left\{0,1,\ldots ,\left(N_{s}-1\right)\right\},c_{i}=\left\{0,1,\ldots ,\left(N_{c}-1\right)\right\}.$$
The time ($t_{s_{i},c_{i}}$) and modulation frequency ($f_{s_{i},c_{i}}$) can be expressed as a two-dimensional (2D) matrix for the ADC samples and chirps, as follows:(16)$$t_{s_{i},c_{i}}=\frac{s_{i}\cdot T_{ADC}}{N_{s}-1}+c_{i}\cdot PRI,$$
(17)$$f_{s_{i},c_{i}}=f_{s}+\frac{s_{i}\cdot BW}{N_{s}-1},$$
where $T_{ADC}$ indicates the ADC sampling time, *PRI* represents the pulse repetition interval, $f_{s}$ denotes the starting frequency, and *BW* indicates the bandwidth (Table 2). Additionally, the delay time induced by the target distance ($\tau_{r}$) and Doppler ($\tau_{d}$) can be expressed as
(18)$$\tau_{r}=\frac{{\left[R_{tx}\right]}_{n}+{\left[R_{rx}\right]}_{m}}{c},$$
(19)$$\tau_{d}=\frac{2V_{r}\cdot t_{s_{i},c_{i}}}{c},$$
where *c* denotes the speed of light, *n* denotes the Rx channel, *m* indicates the Tx channel, and $V_{r}$ represents the radial velocity, as defined by Equation (8). Additionally, each signal in the Tx channel (${Tx}_{s_{i},c_{i}}$) and the Rx channel (${Rx}_{s_{i},c_{i}}$) can be expressed as follows:(20)$${\left[{Tx}_{s_{i},c_{i}}\right]}_{n}={\left[cos\left(2\pi \cdot f_{s_{i},c_{i}}\cdot t_{s_{i},c_{i}}\right)\right]}_{n},$$
(21)$${\left[{Rx}_{s_{i},c_{i}}\right]}_{n,m}={\left[cos\left(2\pi \cdot f_{s_{i},c_{i}}\cdot \left(t_{s_{i},c_{i}}+\tau_{r}+\tau_{d}\right)\right)\right]}_{n,m}.$$
Finally, the Tx signal ($S_{tx}$) and Rx signal for each channel (${\left[S_{rx}\right]}_{m}$) can be obtained as
(22)$$S_{tx}=\sum_{k=1}^{n}{\left[{Tx}_{s_{i},c_{i}}\right]}_{k},$$
(23)$${\left[S_{rx}\right]}_{m}=\sum_{k=1}^{n}{\left[{Rx}_{s_{i},c_{i}}\right]}_{k,m}.$$
Obtaining the ADC signal using a mixed signal is identical to that indicated in Equation (14), and the RV spectrum can be obtained using the 2D fast Fourier transform of the ADC signal [33]. Figure 7 outlines the simulation method.

#### 3.2.2. Simulation Results

Figure 8 presents a comparison of the simulation results described in Section 3.2.1 with the results obtained from the measured data. Figure 8a presents the measured results obtained using the automotive radar system, and Figure 8b shows the simulation results with respect to the targets, which are similar to the measurement results. The dashed red line indicates the left continuous and stationary objects (the left guardrail), while the solid red line represents the right continuous and stationary objects (the right guardrail). This result indicates adequate agreement between the simulation and measurement results, confirming that the simulation data can be used to ensure diversity in datasets and secure insufficient datasets from the measurement results. 

We simulated and extracted various datasets by assuming a fixed object with a continuous uniform distribution over a certain range for every simulation without being biased toward specific cases. The distributions can be indicated as
(24)$$U\left(a,b\right)=\left\{\begin{array}{cc} \frac{1}{b-a}, & \;\;a\leq x\leq b\\ 0, & otherwise \end{array}\right..$$

Based on the coordinates indicated in Figure 4a, Table 3 lists the continuous uniform distribution range of the stationary object sets. Here, X (the lateral direction from the vehicle) ranges from −15 m to +15 m, and Y (the longitudinal direction from the vehicle) begins at 1 m–3 m and ends at 70 m–90 m at 1 m spacing. The number of stationary object sets ranges from one to two. For example, when +10 m is selected from the x position (−15 m~+15 m) by the continuous uniform distribution, 2 m is selected from the Y start position (1 m~3 m), and 80 m is selected from the Y end position (70 m~90 m), a set of stationary objects with X of 3 m and a spacing of 1 m between 2 m and 80 m of Y is generated. 

Moreover, when the number of stationary object sets is two, two different sets of stationary objects are generated by a continuous uniform distribution.

As mentioned in Section 3.1.3, the objective of this study is to estimate the ego-vehicle speed using the stationary target of the RV spectrum. Therefore, to secure robustness for training, extraneous objects such as moving vehicles, which interfere with the estimation of the ego-vehicle speed on the road, were created in the range presented in Table 4. Similar to the stationary objects, the extraneous objects were generated with X ranging from −10 m to +10 m, Y ranging from 1 m to 90 m, and the radial velocity ranging from −23 m/s to +23 m/s based on the coordinates depicted in Figure 4a. In total, 20 to 25 extraneous objects were created, and the continuous uniform distribution was used in the same way as that presented in Equation (24). For example, when 0 m is selected from x (−10 m~+10 m) by the continuous uniform distribution, 40 m is selected from Y (1 m~90 m), and 5 m/s is selected from radial velocity (−23 m/s~+23 m/s), an extraneous object with a radial velocity of 5 m/s is created at the location where X is 0 m and Y is 40 m. Additionally, when 22 extraneous objects are selected, 22 different extraneous objects are created by a continuous uniform distribution. Finally, we used 73,447 and 60,000 frames of measurement and simulation data, respectively, to train the ego-vehicle speed (Table 5).

### 3.3. Implementation of CNN

#### 3.3.1. Preprocessing

Preprocessing was performed to select and reprocess the data to prevent deviations from the intended objective or a decline in network performance. Initially, data with an ego-vehicle speed below 30 km/h (8.3 m/s) were filtered to increase the probability of straight-driving conditions; the ego-vehicle speed was determined using either CAN or GPS data. Subsequently, 40 frames were windowed, and the frames that exceeded a heading angle of 0.5° were excluded. The GPS, yaw rate, or vehicle radius of curvature was utilized based on the heading angle. As the ego-vehicle speed was estimated regardless of the angular performance of the radar, the RV spectrum of each channel was combined to minimize Gaussian noise with non-coherent characteristics while simultaneously increasing the signal-to-noise ratio of the objects, known as non-coherent summation of RV spectrum. As the simulation data were generated under straight driving conditions, only the RV spectrum was combined during preprocessing. Figure 9 illustrates a flowchart of the preprocessing of the measurement and simulation data.

#### 3.3.2. Network Architecture Considerations

In Equation (5), the stationary target angle (θ) ranges from −90° to +90°. Consequently, the radial velocity ($V_{r}$) ranges from 0 m/s to the negative ego-vehicle speed (${-V}_{ego}$). Therefore, only half of the non-coherent summation of the RV spectrum described in Section 3.3.1, which contains the information of the ego-vehicle speed, was used in the network to improve its efficiency. For instance, the velocity in Figure 8 ranges from −24.27 m/s to +24.27 m/s, yet only the data from −24.27 m/s to 0 m/s are utilized, which is satisfactory. Therefore, the size of the RV spectrum input into the network was 128 × 65.

Table 6 presents the experimental results, which demonstrate that a non-uniform stride (2 × 1) outperforms a uniform stride (2 × 2) in estimating the ego-vehicle speed despite the number of parameters being reduced by 1/3. This implies that no information loss occurs for the 128 distance indices in the case of data compression of continuous stationary objects; however, for the 65 velocity indices, information loss occurs when a uniform stride (2 × 2) is used. Therefore, a non-uniform stride (2 × 1) must be used for effective data compression while minimizing the loss of velocity information.

Table 7 summarizes the impact of the kernel size of the CNN on the network performance. When the kernel size is three, a performance similar to that of a kernel size of five is achieved while reducing the number of parameters by 1/3. This implies that the velocity information can be contained within the convolutional filter (3 × 3). For example, using the convolutional filter (3 × 3), stationary objects within a distance of 50 m to 90 m can contain information of three velocity indices, as shown in Figure 8. Therefore, expanding the kernel size does not significantly impact the network performance. In other words, using a minimum kernel size is reasonable when the number of parameters increases rapidly.

#### 3.3.3. Network Architecture

The proposed CNN-based AVSD Net model estimates the ego-vehicle speed in automotive radar systems. Table 8 lists the channel sizes, parameters, and floating-point operations (FLOPs) of six AVSD Net models that received 128 × 65 input data and comprised seven AVSD blocks, a fully connected layer, and Softmax. Figure 10 depicts the composition of the AVSD block. The convolutional layer of the AVSD block comprised a 3 × 3 convolutional filter, a 2 × 1 non-uniform stride, and replication padding. Additionally, the convolutional layer applied different numbers of input/output channels for the six AVSD Net models listed in Table 8. Batch normalization was applied to the model after the convolutional layer and before the activation function [34], and stochastic gradient descent with a mini-batch size of 64 was used. The output obtained after Softmax contained encoded bit information that compressed the velocity information of the stationary objects. The velocity index of the ego-vehicle speed ranging from 0 to 65 was obtained by decoding this bit information via binary-to-decimal conversion. Finally, the ego-vehicle speed was estimated by multiplying the obtained velocity index and the velocity resolution ($V_{res}$) presented in Table 2. Figure 11 and Figure 12 illustrate the architectures of the AVSD Net-84k and AVSD Net-717 models, respectively.

#### 3.3.4. Postprocessing

The ego-vehicle speed estimated using the AVSD Net model introduced in Section 3.3.3 may contain abnormal values. Therefore, the abnormal values must be suppressed via postprocessing to implement a robust system. 

Initially, an exception was processed when the deviation between the current and previous heading angles of the vehicle exceeded the reference value. The yaw rate or the radius of curvature of the vehicle could be used rather than the heading angle. Subsequently, the real-time speed ratio between the real-time ego-vehicle speed received from the CAN of the vehicle and the ego-vehicle speed estimated using AVSD Net was calculated as follows:(25)$$Realtime\;Speed\;Ratio=\frac{Estimated\;Egovehicle\;Speed\;from\;AVSD\;Net}{Realtime\;Egovehicle\;Speed\;from\;CAN}.$$
Finally, abnormally high real-time speed ratio values were eliminated using the speed ratio stored in the electrically erasable programmable read-only memory (EEPROM). A sliding window was considered over time to obtain the real-time speed ratios after eliminating the abnormal values. Subsequently, the median value was extracted, and the final extracted speed ratio was stored in EEPROM. Figure 13 depicts a flowchart of postprocessing.

In summary, the ego-vehicle speed correction in an automotive radar system was performed in real time using the speed ratio stored in the EEPROM. Figure 14 illustrates the preprocessing, network modeling, and postprocessing steps.

#### 3.3.5. Comparison of AVSD Net Models

When applying the AVSD Net model to automotive radar systems, both processing time and memory availability need to be considered. Table 9 presents a comparison of the proposed AVSD Net in this paper for ego-vehicle speed estimation. Since the automotive radar system performs serial processing based on the microcontroller unit, the comparison of processing time was performed using CPU-based serial processing. The model size was compared to the volume of data that each model and its final parameters contained. The processing times of the AVSD Net-717, the simplest model, and the AVSD Net-84k, the most complex model, were 1.33 ms and 2.5 ms, respectively, indicating a difference of approximately twice. Additionally, the model size was 26 and 355 kilobytes, indicating a difference of approximately 14 times.

## 4. Results and Discussion

The datasets used for training were simulation datasets 1 and 2, measurement dataset 1, and half of measurement dataset 2 (Table 5). The remaining half of measurement dataset 2 was used for testing. Each model was trained for $2\times 10^{4}$ iterations through 20 epochs. Figure 15 illustrates the AVSD Net model results of training and testing error rates with respect to GPS data, with Table 10 listing the final error rates. The AVSD Net-84k model, comprising the highest number of parameters, exhibits training and testing error rates of 0.02% and 5.29%, respectively. Conversely, the AVSD Net-717 model, which is a simple model with the least number of parameters, exhibits training and testing error rates of 1.45% and 7.39%, respectively. Overall, the training and testing error rates decreased slightly in each model as the number of parameters increased. These results are similar to those reported in other related studies based on CNN [35,36,37]. This result can be attributed to training with increasing weights for stationary objects located at long distances as the number of channels increases in each AVSD Net model.

The obtained results imply that the selection of a model depends on the hardware performance of the automotive radar system. If automotive radar systems are expensive and exhibit high performance, the AVSD Net-84k model is preferred. However, most automotive radar system manufacturers adopt minimal and optimized hardware performances to increase profits. Therefore, to consider the performance of an embedded system with optimized hardware, the ego-vehicle speed correction must be optimized via postprocessing using a simplified model.

The postprocessing verification results, as explained in Section 3.3.4 for each model with the remaining half of measurement dataset 2, are illustrated in Figure 16. The blue line in the figure represents the postprocessing results before the application of the sliding window to the real-time speed ratio described in Equation (25). In this scenario, the error-limited filter restricts the change in the heading angle to less than 0.5° and limits the difference in the speed ratio stored in the EEPROM to 0.05. The orange line indicates the postprocessing results after the application of the sliding window to the real-time speed ratio with a window size of 150; the initial absence of data in the orange line denotes the interval used for securing the required window size of 150. Table 11 summarizes the postprocessing results of each model. The overall training error rate increases for the dataset; however, the speed ratio for all models converges to a value of approximately 1.018 after postprocessing. This value is close to the speed ratio of 1.0183, which was obtained from GPS measurements (Section 3.1.3). The standard deviation of the speed ratio is negligible.

The obtained results confirm that all six models designed for ego-vehicle speed estimation can achieve equivalent performance levels via postprocessing. Ideally, a model with the maximum number of parameters must be selected; however, selecting the AVSD Net-717 model with the minimum number of parameters can be reasonable considering the hardware performance. This approach ensures optimal performance without requiring a complex model architecture.

Figure 17 compares the results when the window size of the AVSD Net-171 model is changed. The orange line that represents the result of postprocessing appears stable for window sizes of 150 and 100; however, an abnormal value is observed when the window sizes are 50 and 25. Table 12 summarizes the anomalous findings depicted in Figure 17. The rates of abnormal value are 2.65% and 8.55% for window sizes of 50 and 25, respectively, and the standard deviation of the speed ratio increases rapidly. Figure 18 depicts the rate of abnormal value with respect to the change in window size in this dataset. Here, the rate of abnormal value reaches zero when the window size is 81 or higher. However, considering the possibility of unexpected scenarios, a margin for error must be applied. Therefore, a window size that ranges from 100 to 150 can be considered appropriate. This result demonstrates the applicability of a model with minimal complexity to automotive radar systems to perform ego-vehicle speed correction via postprocessing optimization.

## 5. Conclusions

In this study, we designed six CNN-based models specifically for ego-vehicle speed estimation. The training and testing of each model indicated that both training and testing error rates tended to decrease with the increase in the number of parameters. However, considering the limitations of the embedded system, the AVSD Net-717 model, which comprised the minimum number of parameters, was deemed optimal. Therefore, we examined the AVSD Net-717 model by adjusting the window size to achieve satisfactory performance via postprocessing optimization. Our analysis indicated that the optimal performance and minimal complexity methods of the AVSD Net-717 model can be derived by postprocessing with a window size ranging from 100 to 150. The final speed ratio obtained after postprocessing is approximately 1.018. This value is close to the speed ratio obtained from GPS measurements (Section 3.1.3). However, there exists a trade-off between model complexity and the initial correction time of the ego-vehicle speed. For instance, a simple model can be used in automotive radar systems with poor hardware performance; however, the initial correction time of the ego-vehicle speed increases because of the large window size. Conversely, high-performing automotive radar systems that can handle models with high complexity use small window sizes, thereby reducing the initial correction time of the ego-vehicle speed. This trade-off relationship should be appropriately adjusted based on the requirements of the automotive radar system.

Since the ADC data of the automotive radar system used in this study were acquired as one frame at a cycle of 0.05 s, and the data size per frame is 0.5 MB, if data are acquired in various environments considering all situations, difficulties would arise owing to the large capacity and excessive acquisition time. Therefore, we efficiently simulated various environments and used them for training. However, simulation data do not provide all data for unpredicted environments. Although this study has opened up the possibility of applying deep learning to estimate the ego-vehicle speed, future research should acquire data from a wider range of environments and scenarios and prepare the data for training.

It is also important to note that current automotive radar systems do not utilize INS/GNSS-based filtering and fusion techniques, which are more common in navigation systems. However, techniques such as the double-channel sequential probability ratio test [38], hypothesis test-constrained robust Kalman filter [39], and set-membership-based hybrid Kalman filter [40] have proven their effectiveness in managing sensor fusion and uncertainty in INS/GNSS applications. These methods, including the adaptive and robust cubature Kalman filter [41,42,43], could potentially be adapted to enhance radar systems, particularly in handling complex environments and sensor noise. As future research progresses, exploring the integration of these INS/GNSS-based methods into radar systems could provide an effective approach to further improve real-time performance and robustness in unpredictable scenarios.

## Figures and Tables

**Figure 1 sensors-24-06409-f001:**
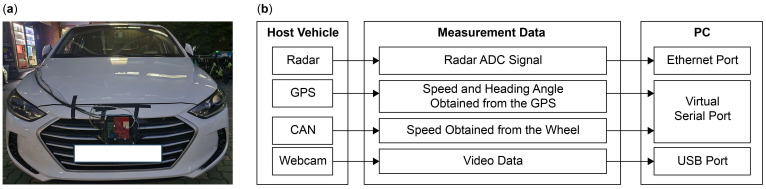
Configuration of the vehicle and radar system: (**a**) photograph; (**b**) schematic.

**Figure 2 sensors-24-06409-f002:**
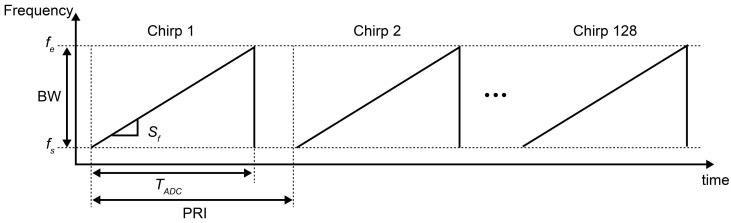
Waveform of the automotive radar system.

**Figure 3 sensors-24-06409-f003:**
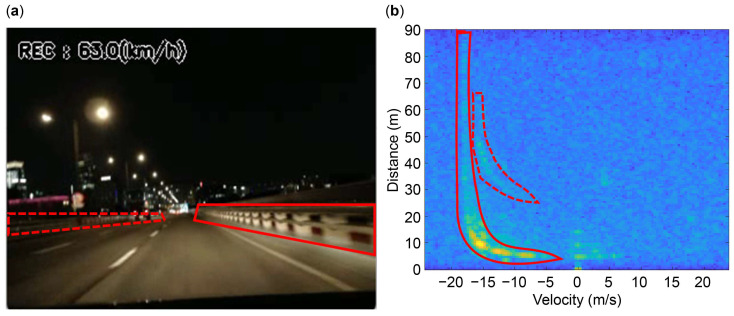
Measurement results at an ego-vehicle speed of 17.5 m/s. The dashed and solid red lines represent the left and right guardrails, respectively: (**a**) Video acquired from the webcam; (**b**) range-velocity (RV) spectrum data acquired from the automotive radar.

**Figure 4 sensors-24-06409-f004:**
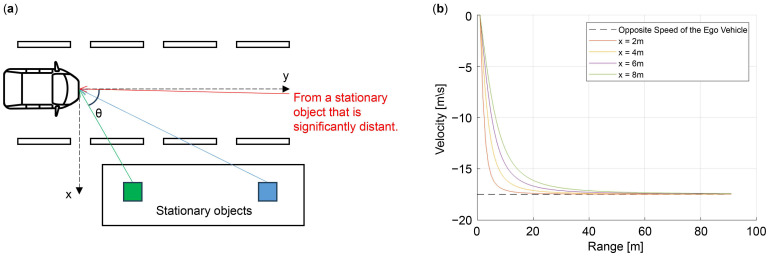
Variation in the radial velocity based on the position of objects: (**a**) automotive radar coordinate system; (**b**) radial velocity according to position x of the stationary objects.

**Figure 5 sensors-24-06409-f005:**
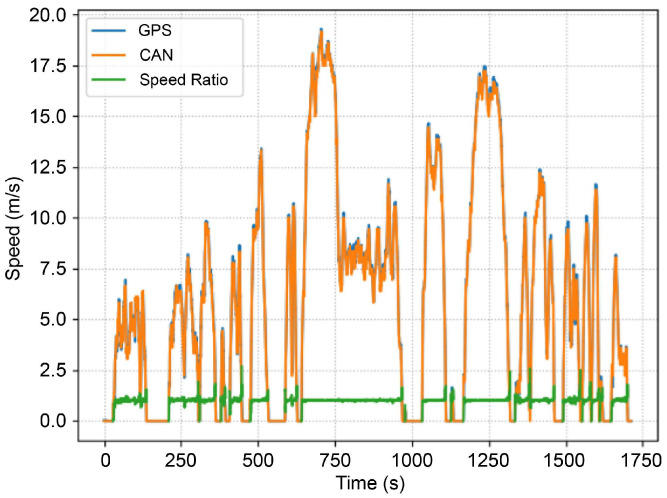
Ego-vehicle speed obtained from the global positioning system (GPS) and controller area network (CAN).

**Figure 6 sensors-24-06409-f006:**
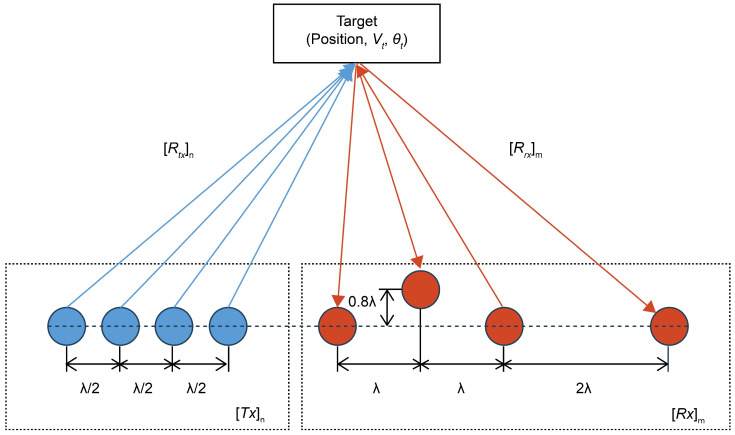
Position of the target and antenna.

**Figure 7 sensors-24-06409-f007:**
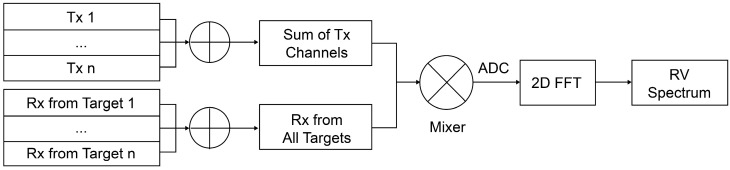
Schematic of the simulation method.

**Figure 8 sensors-24-06409-f008:**
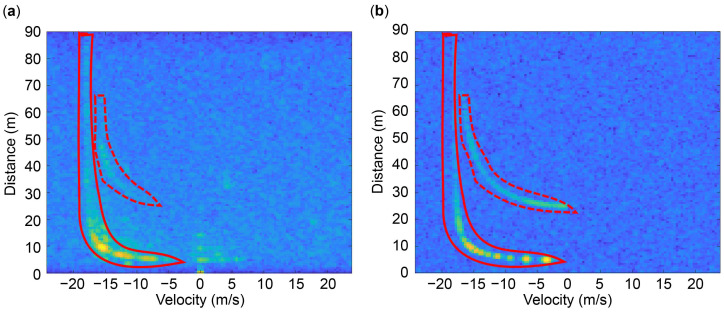
Comparison of results: (**a**) measurement data; (**b**) simulation data.

**Figure 9 sensors-24-06409-f009:**
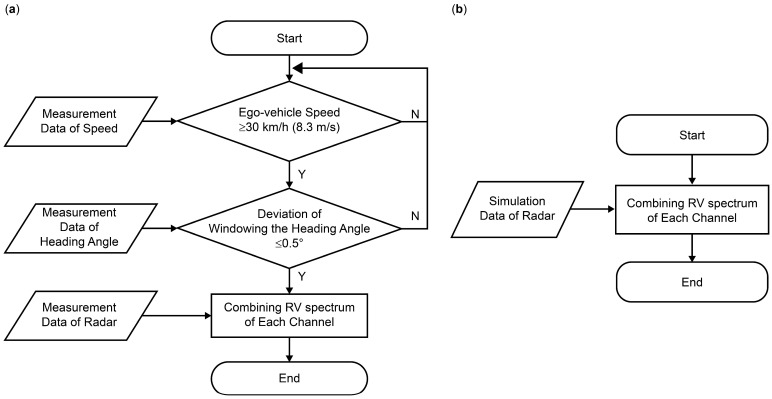
Preprocessing flowchart: (**a**) measurement data; (**b**) simulation data.

**Figure 10 sensors-24-06409-f010:**
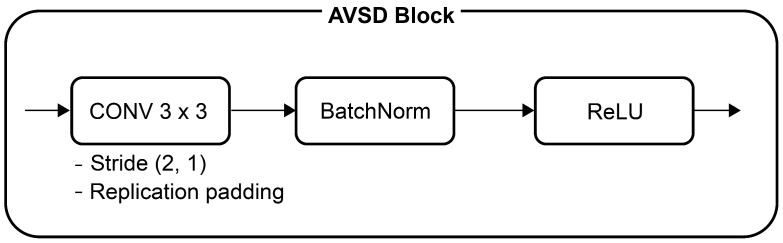
Automotive radar-based ego-vehicle speed detection (AVSD) block.

**Figure 11 sensors-24-06409-f011:**
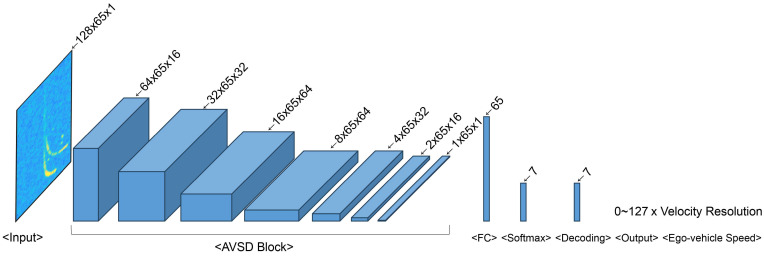
Architecture of the automotive radar-based ego-vehicle speed detection network (AVSD Net)-84k.

**Figure 12 sensors-24-06409-f012:**
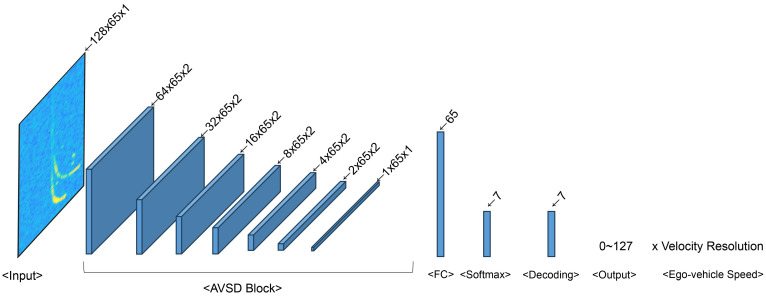
Architecture of AVSD Net-717.

**Figure 13 sensors-24-06409-f013:**
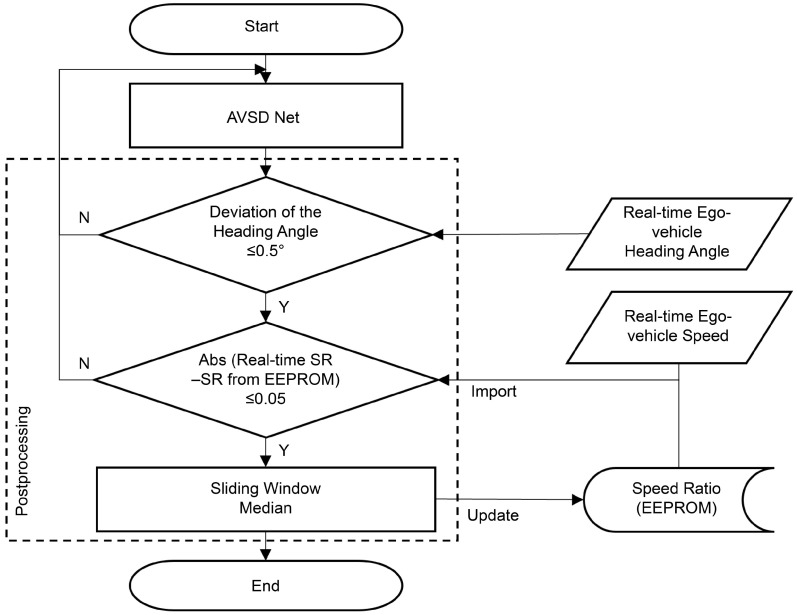
Postprocessing flowchart.

**Figure 14 sensors-24-06409-f014:**
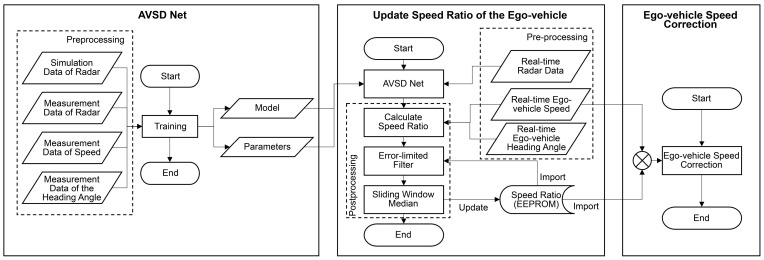
Method followed for ego-vehicle speed correction.

**Figure 15 sensors-24-06409-f015:**
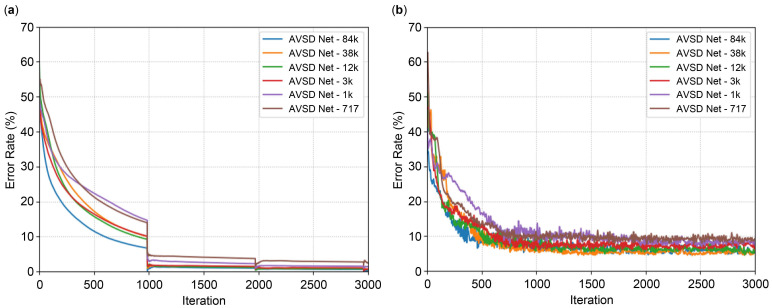
Results obtained from the AVSD Net model: (**a**) training error rate; (**b**) testing error rate.

**Figure 16 sensors-24-06409-f016:**
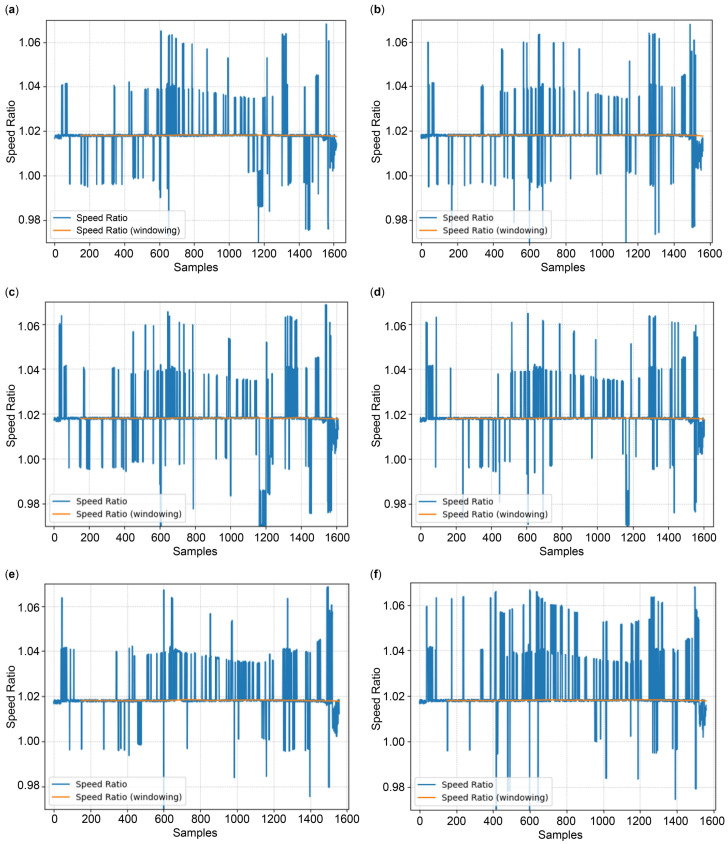
Speed ratio results after postprocessing: (**a**) AVSD Net-84k; (**b**) AVSD Net-38k; (**c**) AVSD Net-12k; (**d**) AVSD Net-3k; (**e**) AVSD Net-1k; (**f**) AVSD Net-717.

**Figure 17 sensors-24-06409-f017:**
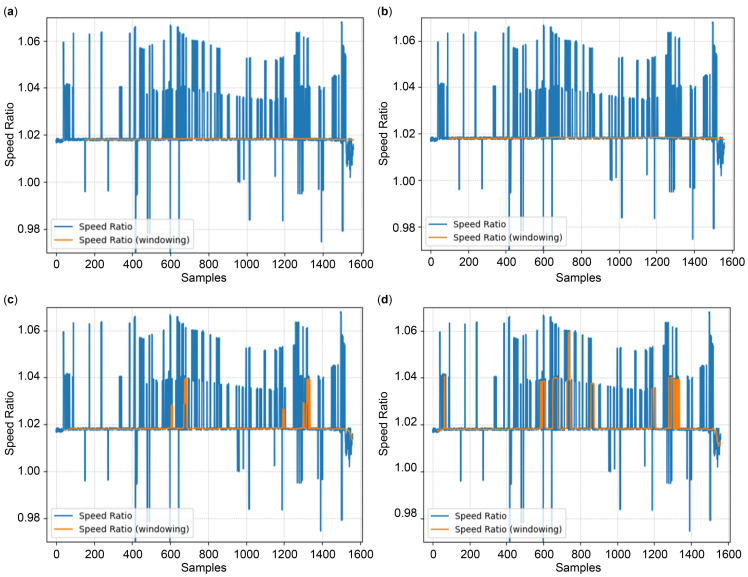
Speed ratio results after postprocessing of AVSD Net-717 for various window sizes: (**a**) 150; (**b**) 100; (**c**) 50; (**d**) 25.

**Figure 18 sensors-24-06409-f018:**
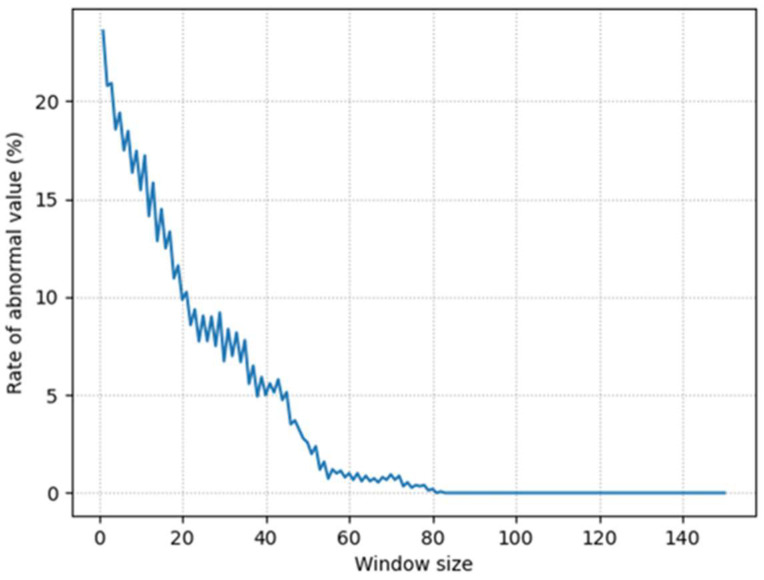
Rate of abnormal value with respect to window size.

**Table 1 sensors-24-06409-t001:** Constraints associated with the estimation of ego-vehicle speed.

Study	Mounting Angle	Angular Performance	Downward or Wide Beam	Complexity
ODR	Dependent	Dependent	Unnecessary	Simple
Hough transform	Dependent	Dependent	Unnecessary	Simple
Elevated objects	Dependent	Dependent	Necessary	Simple
Ground backscattering	Dependent	Dependent	Necessary	Simple
RANSAC	Independent	Dependent	Unnecessary	Simple
CVNN	Independent	Independent	Unnecessary	Complex

**Table 2 sensors-24-06409-t002:** Specifications of the waveform of the automotive radar system.

Parameter	Value	Parameter	Value
$\mathrm{Start}\;\mathrm{frequency}\;(f_{s}$)	77.0991 GHz	$\mathrm{Number}\;\mathrm{of}\;\mathrm{ADC}\;\mathrm{samples}\;(N_{s}$)	256
$\mathrm{End}\;\mathrm{frequency}\;(f_{e}$)	77.3104 GHz	$\mathrm{Number}\;\mathrm{of}\;\mathrm{chirps}\;(N_{c}$)	128
Bandwidth (BW)	211.3536 MHz	$\mathrm{Frame}\;\mathrm{duration}\;(T_{frm}$)	50 ms
Sample rate (SR)	20 MHz	$\mathrm{Range}\;\mathrm{resolution}\;(R_{res}$)	0.7092 m
$\mathrm{ADC}\;\mathrm{sampling}\;\mathrm{time}\;(T_{ADC}$)	12.8 µs	$\mathrm{Maximum}\;\mathrm{range}\;(R_{max}$)	90.7802 m
$\mathrm{Frequency}\;\mathrm{slope}\;(S_{f}$)	16.512 MHz/µs	$\mathrm{Velocity}\;\mathrm{resolution}\;(V_{res}$)	0.3792 m/s
Pulse repetition interval (PRI)	40 µs	$\mathrm{Maximum}\;\mathrm{velocity}\;(V_{max}$)	24.2693 m/s

**Table 3 sensors-24-06409-t003:** Continuous uniform distribution range of stationary object sets.

Range	X (m)	Y Start (m)	Y End (m)	Y Direction Spacing (m)	Number of Stationary Object Sets
a	−15	1	70	1	1
b	+15	3	90	-	2

**Table 4 sensors-24-06409-t004:** Continuous uniform distribution range of extraneous objects.

Range	X (m)	Y (m)	Radial Velocity (m/s)	Number of Extraneous Objects
a	−10	1	−23	20
b	+10	90	+23	25

**Table 5 sensors-24-06409-t005:** Number of measurement and simulation datasets.

Datasets	Measurement Data	Simulation Data
Dataset 1	34,538	30,000
Dataset 2	38,909	30,000
Total	73,447	60,000

**Table 6 sensors-24-06409-t006:** Comparison of uniform stride (2 × 2) and non-uniform stride (2 × 1).

Layer Name	Uniform Stride (2 × 2)				Non-Uniform Stride (2 × 1)			
	Input Size	Output Size	Channels (Input)	Channels (Output)	Input Size	Output Size	Channels (Input)	Channels (Output)
Input	-	128 × 65	-	1	-	128 × 65	-	1
Conv 1	128 × 65	64 × 33	1	8	128 × 65	64 × 65	1	8
Conv 2	64 × 33	32 × 17	8	16	64 × 65	32 × 65	8	8
Conv 3	32 × 17	17 × 9	16	16	32 × 65	16 × 65	8	8
Conv 4	16 × 9	8 × 5	16	16	16 × 65	8 × 65	8	8
Conv 5	8 × 5	4 × 3	16	16	8 × 65	4 × 65	8	8
Conv 6	4 × 3	2 × 2	16	16	4 × 65	2 × 65	8	8
Conv 7	-	-	-	-	2 × 65	1 × 65	8	1
FC	64	7	-	-	65	7	-	-
Params	11,306				3633			
Error	7.09%				4.26%			

**Table 7 sensors-24-06409-t007:** Comparison of kernel sizes of (5 × 5) and (3 × 3).

Layer Name	Input Size	Output Size	Kernel Size (5 × 5)		Kernel Size (3 × 3)	
			Channels (Input)	Channels (Output)	Channels (Input)	Channels (Output)
Input	-	128 × 65	-	1	-	1
Conv 1	128 × 65	64 × 65	1	16	1	16
Conv 2	64 × 65	32 × 65	16	32	16	32
Conv 3	32 × 65	16 × 65	32	32	32	32
Conv 4	16 × 65	8 × 65	32	32	32	32
Conv 5	8 × 65	4 × 65	32	32	32	32
Conv 6	4 × 65	2 × 65	32	16	32	16
Conv 7	2 × 65	1 × 65	16	1	16	1
FC	65	7	-	-	-	-
Params	-		104,145		38,097	
Error	-		4.22%		3.99%	

**Table 8 sensors-24-06409-t008:** Channels and parameters of the automotive radar-based ego-vehicle speed detection network (AVSD Net) model.

Layer Name	Input Size	Output Size	AVSD NET-84k		AVSD NET-38k		AVSD NET-12k		AVSD NET-3k		AVSD NET-1k		AVSD NET-717	
			Channels		Channels		Channels		Channels		Channels		Channels	
			In	Out	In	Out	In	Out	In	Out	In	Out	In	Out
Input	-	128 × 65	-	1	-	1	-	1	-	1	-	1	-	1
Conv 1	128 × 65	64 × 65	1	16	1	16	1	16	1	8	1	4	1	2
Conv 2	64 × 65	32 × 65	16	32	16	32	16	16	8	8	4	4	2	2
Conv 3	32 × 65	16 × 65	32	64	32	32	16	16	8	8	4	4	2	2
Conv 4	16 × 65	8 × 65	64	64	32	32	16	16	8	8	4	4	2	2
Conv 5	8 × 65	4 × 65	64	32	32	32	16	16	8	8	4	4	2	2
Conv 6	4 × 65	2 × 65	32	16	32	16	16	16	8	8	4	4	2	2
Conv 7	2 × 65	1 × 65	16	1	16	1	16	1	8	1	4	1	2	1
FC	65	7	-	-	-	-	-	-	-	-	-	-	-	-
Softmax	7	7	-	-	-	-	-	-	-	-	-	-	-	-
Parameters	-		84,369		38,097		12,561		3633		1329		717	
FLOPs	-		55,141 k		28,533 k		10,550 k		2954 k		897 k		304 k	

**Table 9 sensors-24-06409-t009:** Comparison of processing time and model size of AVSD Net models.

Model	Processing Time in CPU	Model Size
AVSD Net-84k	2.50 ms	355 kilobytes
AVSD Net-38k	2.34 ms	174 kilobytes
AVSD Net-12k	1.99 ms	74 kilobytes
AVSD Net-3k	1.71 ms	37 kilobytes
AVSD Net-1k	1.44 ms	29 kilobytes
AVSD Net-717	1.33 ms	26 kilobytes

**Table 10 sensors-24-06409-t010:** Error rates of training and testing.

Model	Number of Parameters	Training Error Rate	Testing Error Rate
AVSD Net-84k	84,369	0.02%	5.29%
AVSD Net-38k	38,097	0.10%	5.01%
AVSD Net-12k	12,561	0.17%	6.09%
AVSD Net-3k	3633	0.32%	5.89%
AVSD Net-1k	1329	0.64%	7.31%
AVSD Net-717	717	1.45%	7.39%

**Table 11 sensors-24-06409-t011:** Testing error rate and speed ratio after postprocessing.

Model	Training Error Rate	Mean of Speed Ratio	Standard Deviation of Speed Ratio
AVSD Net-84k	7.00%	1.01813	$10.19\times 10^{-5}$
AVSD Net-38k	9.47%	1.01816	$7.85\times 10^{-5}$
AVSD Net-12k	9.29%	1.01813	$10.91\times 10^{-5}$
AVSD Net-3k	7.71%	1.01817	$9.37\times 10^{-5}$
AVSD Net-1k	8.82%	1.01826	$13.63\times 10^{-5}$
AVSD Net-717	14.94%	1.01823	$12.12\times 10^{-5}$

**Table 12 sensors-24-06409-t012:** Speed ratio after the application of sliding window over the entire measurement dataset 2.

Window Size	Mean of Speed Ratio	Standard Deviation of Speed Ratio	Rate of Abnormal Value
150	1.01813	$12.12\times 10^{-5}$	0.00%
100	1.01822	$14.40\times 10^{-5}$	0.00%
50	1.01869	$291.50\times 10^{-5}$	2.65%
25	1.01994	$618.78\times 10^{-5}$	8.55%

## Data Availability

The datasets presented in this article are not readily available due to proprietary concerns. Requests to access the datasets should be directed to the corresponding author.

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
