# Peer review of "Ego-Vehicle Speed Correction for Automotive Radar Systems Using Convolutional Neural Networks"

_sensors, 2024, doi:10.3390/s24196409_

Round 1

Reviewer 1 Report

Comments and Suggestions for Authors

In the current study authors attempted to introduce an automotive radar-based  vehicle speed detection  model using convolutional neural networks for vehicle speed estimation. While the paper was well-written and interesting, the following is a suggestion for the authors:

1. Authors should add, in the methods section, information about the processing speed of the computer used and the time taken to run the simulations. The readership may want to know for the purposes of performance benchmarking, reproducibility, giving insights regarding feasibility, identification of software constraints and practical applications.

Author Response

Attached separately. 

Reviewer 2 Report

Comments and Suggestions for Authors

General comments.

The work is based on the SW simulation of FMCW automotive radar signals with chirp modulation assuming fixed targets and the application of convolutional networks to the simulations and true radar measurements to estimate the vehicle speed.

The authors should include some statements and references regarding the specifications of precision and accuracy of the speed measurements required by the different automotive automatic aid systems. This is to understand what needs are needed and whether what they have developed is sufficient.

If I understood well, and the eq (6) and (25) are correct (see specific comments #8, #23 and #25 below),  the overall “average” results are that the AVSD speed estimate is lower than that of CAN (speed ratio CAN/AVSD is around 1.02). Such overall result appears to be reliable taking into account that generally vehicle speedometers are calibrated by imposing a positive bias for safety reasons. In other words, the speed shown is always a little higher than the real one. Furthermore, increasing the complexity of the CNN does not show any significant improvement in performance.

The content is interesting, but some major and minor revisions are required.

Below the list of questions, comments and suggestions.

1.       In the conclusion section no data about the performance of the Ego-vehicle Speed Correction. Please summarize here the results (i.e the average speed ratio AVSD/CAN is xxxx  and GPS/CAN is yyyy)

2.       line 152-153 I suggest to move such lines at the beginning of sec 3.1.2  in order to have an unique parameter set definition for the FMCW radar sensor before their use in the text.

3.       Eq. 1 – and Eq. 2 delete the apostrophe

4.       Eq. 2 and lines 142-146. The Maximum non ambiguous range Rmax (in FMCW radar as the TI AWR2944 where the sampling is in baseband and not at intermediate freq.) using the parameters in table 1 is given by (c*TADC*SR)/(4*BW) that gives (3e8*12.8e-6*20e6)/(4*211.3e6)=90.9 m, therefore the maximum range value reported in line 147 is  correct.

5.       It easier add a division per 2 to the Eq. 2 and remove the sentence about I and Q sampling.

6.       Lines 161 and 165 – Is the EGO speed of 17.5 m/s the one measured by the CAN? Please, add a comment.

7.       Fig.5 – maybe adding a plot with the absolute and relative speed difference help the readers

8.       Lines 200-201. A speed ratio greater than 1 means that the GPS speed measurements are generally greater that those of CAN. I would have thought of obtaining the opposite (speed ratio <1) since the speed measurement of vehicle speedometers generally provide values ​​with a positive bias compared to the true speed value for safety reasons. Authors use the term error for the speed measurement difference between GPS and CAN. Since we don’t know the true independent speed, I suggest not using the term error but another term (i.e. Delta V).

9.       Eq 9 – I don’t’ understand its use (also wrong math notation with the symbol t for both the integral upper limit and integration variable dt). Simply assess that we assume FMCW with chirp modulation (linear variation of frequency VS time) where f(t)=fc+DF*t .

10.   Eq 10. Missing t and better add an amplitude A as Stx(t)=A cos (2*pi*f(t)*t).

11.   Line 258 - Not everyone is aware that with the double FFT on the FMCW chirps frames it is possible to obtain range speed maps. I suggest adding at least one reference describing the double FFT on FMCW radar signals.

12.   Line 275 - The term uniform distribution to define the guard rail is inappropriate. It is not a random variable. I don’t understand the meaning of the function f(x). Please rewrite this section.

13.   Line 282- Table 3 is not clear.  Parameter a,b (range?) are associated with a single guard rail? What does spacing mean? Y start?

14.   Line 288  – Noise objects is inappropriate, they are vehicles! Using noise for radar target signal is confusing. Ok for accounting such signals in the simulation but change the terms.

15.   Line 298- What do dataset1 and dataset2 mean? Which are the differences between them? CNN training, test, check. Validation purpose?? Please add description.

16.   I understand that you have a scenario from -15 to 15 in x direction and 0 – 90 in y direction. You have one left (long from 1 to 70 ) and one right (long from 3 to 90) guard rails and a set of moving targets in the middle with random assignment of speed (in -23, 23 m/s) and position (1-90) for generating the simulation runs. Please, rewrite the simulation scenario description and review the table 3 and 4.

17.   Do you consider thermal noise on the received signal model? Please, comment on this.

18.   How du you manage the signal amplitude of the guardrail with respect to that of other targets? Please, comment on this.

19.   Do you consider the effect of interfering signals (i.e. other vehicle radar signals, other emitted radio signals)? Please, comment on this.

20.   Lines 326-332 - I don’t understand the question about the uniform – non uniform stride (instead stride, maybe better resolution? Step? Sampling?) and the importance of this.

21.   Line 339 – 341 authors write: Figure 8 confirms that the distant stationary objects are within the three velocity indices. Therefore, expanding the kernel size does not impact the network performance significantly.” Please give more info about the metric for quantifying the network performance.

22.   Figure 8 (a?? b??, both a and b???) is qualitative and therefore referring to the figure to confirm something about resolutions and performance is not appropriate.

23.   Eq 25 - With respect to eq 6 and to uniform the result analysis, move CAN to the denominator.

24.   Line 408 – figure 15. What does “error rate” is? Refer also to comment #21 and define better the error rate parameter. Could you explain the behavior (step) at iteration 1000 in Fig. 15 (a)??

25.   Line 427-430 – pay attention to the speed ratio definition in sec 3.1.3, that is GPS/CAN (eq 6), and that of eg 25 thatis CAN/AVSS, therefore if eq 6 and 25 are correct, the sentence “This value is close to the speed ratio of 1.0183” is not correct. Please check it,

Author Response

Attached separately. 

Reviewer 3 Report

Comments and Suggestions for Authors

   This paper presents an Ego-vehicle Speed Correction method for Automotive Radar Systems Using Convolutional Neural Networks,the topic of this manuscript is attractive.  The experiment in this manuscript are credible and sufficient. Nevertheless, I have some questions as follows:

1.      In section 3, the author mentioned that the radial velocity is less affected by the angle of the objects when stationary objects are far from the automotive radar, and is affected by the angle of the object when stationary objects are close to the automotive radar, in the moving of the vehicle, the distance from the stationary objects to the automotive radar is changeable, so the viewpoint in the abstract that AVSD Net model exhibits characteristics that are independent of the angular performance of the radar system and its mounting angle on the vehicle is too objective. Different circumstances should be classified discussed.

2.Table 1 summarizes the constraints associated with each existing method used for estimating the ego-vehicle speed. Is the performance of the method in this paper better than the methods in Table 1? Can they be compared in the numerical examples?

3.     Fig.15-17 need more explanation. The significance of the results to the proposed algorithm needs to be emphatically analyzed.

Comments on the Quality of English Language

The grammar and the sentence expression should be improved, such as the first sentence in the abstract. 

Author Response

Attached separately. 

Round 2

Reviewer 2 Report

Comments and Suggestions for Authors The authors answered all the questions I asked and the document was improved accordingly by modifying and adding what I requested.  

Limitations, approximations and scientific framework are correctly treated and described

Reviewer 3 Report

Comments and Suggestions for Authors

for comments 2, the performance of the method in this paper should be compared with  some existing method.
